# A Steady State Continuous Flow Chamber
# for the Study of Daytime and Nighttime Chemistry
# under Atmospherically Relevant NO levels

Xuan Zhang [1*], John Ortega [1*], Yuanlong Huang [2], Stephen Shertz [1],
Geoffrey S. Tyndall [1], and John J. Orlando [1]

[1] Atmospheric Chemistry Observation & Modeling Laboratory (ACOM), National Center for Atmospheric Research (NCAR), Boulder, CO, USA

[2] Department of Environmental Science and Engineering, California Institute of Technology, Pasadena, CA, USA

[*] Authors contributed equally to this work.

*Correspondence to*: Xuan Zhang (xuanz@ucar.edu)

**Abstract**
Experiments performed in laboratory chambers have contributed significantly to the
understanding of the fundamental kinetics and mechanisms of the chemical reactions occurring
in the atmosphere. Two chemical regimes, classified as 'high-NO' versus 'zero-NO' conditions,
have been extensively studied in previous chamber experiments. Results derived from these two
chemical scenarios are widely parameterized in chemical transport models to represent key
atmospheric processes in urban and pristine environments. As the anthropogenic $NO_x$ emissions
in the United States have decreased remarkably in the past few decades, the classic 'high-NO'
and 'zero-NO' conditions are no longer applicable to many regions that are constantly impacted
by both polluted and background air masses. We present here the development and
characterization of the NCAR Atmospheric Simulation Chamber, which is operated in steady
state continuous flow mode for the study of atmospheric chemistry under 'intermediate NO'
conditions. This particular chemical regime is characterized by constant sub-ppb levels of NO
and can be created in the chamber by precise control of the inflow NO concentration and the
ratio of chamber mixing to residence timescales. Over the range of conditions achievable in the
chamber, the lifetime of peroxy radicals ($RO_2$), a key intermediate from the atmospheric
degradation of volatile organic compounds (VOCs), can be extended to several minutes, and a
diverse array of reaction pathways, including unimolecular pathways and bimolecular reactions
with NO and $HO_2$, can thus be explored. Characterization experiments under photolytic and dark
conditions were performed and, in conjunction with model predictions, provide a basis for
interpretation of prevailing atmospheric processes in environments with intertwined biogenic and
anthropogenic activities. We demonstrate the proof of concept of the steady state continuous
flow chamber operation through measurements of major first-generation products, methacrolein
(MACR) and methyl vinyl ketone (MVK), from OH- and $NO_3$-initiated oxidation of isoprene.

## 1. Introduction

With the discovery of the role of biogenic volatile organic compounds (BVOCs) in urban photochemical smog (Chameides et al., 1988), the interactions of biogenic emissions with man-made pollution and their subsequent impact on the atmosphere's oxidative capacity and aerosol burden have received extensive studies in the ensuing decades (De Gouw et al., 2005; Ng et al., 2007; Goldstein et al., 2009; Surratt et al., 2010; Rollins et al., 2012; Shilling et al., 2013; Xu et al., 2015). A particular research focus has been understanding the influence of nitrogen oxides ($NO_x = NO + NO_2$) on the atmospheric oxidation cascades of BVOCs, which ultimately generate ozone ($O_3$) and secondary organic aerosols (SOA). Nitrogen oxides alter the distribution of BVOC oxidation products by primarily modulating the fate of peroxy radicals ($RO_2$), a key intermediate produced from the atmospheric degradation of VOCs by major oxidants including OH, $O_3$, and $NO_3$. In the absence of $NO_x$, $RO_2$ reacts predominantly with $HO_2$ radicals yielding organic peroxides and other products, and to a lesser extent, undergoes self/cross-reactions yielding carbonyls, alcohols, and multifunctional species. In the presence of elevated $NO_x$, the dominant fate of $RO_2$ is to react with NO leading to ozone production, and also to organic nitrates. During the night, $RO_2$ also reacts with $NO_3$ which is produced by the reaction between $O_3$ and $NO_2$. In addition, reaction of peroxyacyl radicals ($RC(O)O_2$) with $NO_2$ produces peroxyacyl nitrates that constitute a large reservoir of reactive nitrogen and a potentially important SOA precursor (Singh and Hanst, 1981; Nguyen et al., 2015).

Much of our understanding of the extent to which $NO_x$ mediates the oxidation chemistry of BVOC in the atmosphere has been derived from measurements in laboratory chambers, where two extreme experimental conditions, i.e., 'high-NO' vs. 'zero-NO', were mostly performed to examine the reaction pathways of $RO_2$ radicals (Kroll and Seinfeld, 2008; Orlando and Tyndall, 2012; Ziemann and Atkinson, 2012). Results from these two chemical regimes have been widely incorporated into chemical transport models to represent key atmospheric processes in urban and pristine environments, respectively (Kanakidou et al., 2005). In the actual atmosphere, however, the fate of $RO_2$ radicals is rather more complicated than simply undergoing bimolecular reactions with $NO/HO_2$ as observed under the two extreme chamber conditions. It has been recently revealed that $RO_2$ radicals may undergo an internal H-shift followed by sequential $O_2$ addition, leading to highly oxygenated multifunctional peroxides (Ehn et al., 2014; Jokinen et al., 2015; Kurtén et al., 2015; Kirkby et al., 2016; Zhang et al., 2017). The rate of H-shift largely depends on the thermochemistry of the nascent alkyl radicals and can be reasonably fast, on a time scale of seconds to minutes (Crounse et al., 2013). Further, depending on the stability of the $RO_2$ precursor (alkyl radicals), $RO_2$ radicals may lose $O_2$ in competition with bimolecular reactions with NO, $NO_3$, $RO_2$, and $HO_2$. Recent theoretical and laboratory studies have found that the hydroxy peroxy radical conformers produced from isoprene photooxidation decompose readily to allylic radicals on time scales faster than bimolecular processes under atmospherically relevant $NO/HO_2$ levels (tens to hundreds of parts per trillion by volume). This highly dynamic system leads to formation of distinctly different products that depend on the concentrations of bimolecular reaction partners from those observed in chamber experiments under 'high-NO' and 'zero-NO' conditions (Teng et al., 2017).

Anthropogenic $NO_x$ emissions in the United States have decreased remarkably in the past few decades (EPA, 2014), resulting in significant changes in the degradation mechanisms of BVOCs, especially in regions impacted by both background and polluted air masses such as the Southeastern United States. However, the ultimate fate of peroxy radicals in environments with

sub-ppb NO levels is still poorly constrained, in part due to a lack of consistent measurements
under well controlled conditions. Experimental approaches targeting at a controlled NO level
(sub-ppb to ppb) have been introduced over the years. For outdoor chambers, experiments were
typically performed by exposing a gas mixture of $O_3/NO_x/VOCs$ or $HONO/NO_x/VOCs$ to natural
sunlight (Bloss et al., 2005; Karl et al., 2006). OH radicals were produced either via the
photolysis of ozone and subsequent reaction of $O(^1D)$ with $H_2O$ or directly from the photolysis
of HONO. NO levels ranging from a few hundreds of ppt to a few ppb over the course of several
hours of reactions have been reported. In the absence of any additional supply, NO will be
eventually depleted in a closed chamber environment, and the initial 'moderate-NO' condition
will essentially transfer to the 'zero-NO' condition. For indoor chambers, a 'slow chemistry'
scenario initiated by photolyzing methyl nitrite ($CH_3ONO$) under extremely low UV intensities
as the OH radical source ($J_{CH3ONO} \sim 10^{-5}$ $s^{-1}$) was created to study the autoxidation chemistry of
peroxy radicals produced from isoprene photooxidation (Crounse et al., 2011; Crounse et al.,
2012; Teng et al., 2017). The resulting NO and $HO_2$ mixing ratios are maintained at ~ppt level
($CH_3ONO + O_2 + hv \rightarrow HO_2 + NO + HCHO$) over the course of several hours of reaction, and
the average OH concentration (OH ~ $10^5$ molec $cm^{-3}$) is approximately one order of magnitude
lower than that in the typical daytime ambient atmosphere. Another example relates to a recent
method development in the Potential Aerosol Mass (PAM) flow tube reactor where nitrous oxide
($N_2O$) was used to produce ~ppb level of NO ($O_3 + hv \rightarrow O_2 + O(^1D)$; $O(^1D) + N_2O \rightarrow 2NO$)
(Lambe et al., 2017). Timescales for chemical reactions and gas-particle partitioning are
ultimately limited to the mean residence time (~80 s) of the PAM reactor.

91       An alternative experimental platform to the batch-mode chamber and flow tube reactor
described above is a well-mixed steady-state chamber with continuous feed of reactants and
continuous withdrawal of reactor contents (Shilling et al., 2008). An attribute of the continuous
flow steady state chamber is that, by control of the inlet reactant concentrations and the ratio of
mixing to residence timescales, it is possible to simulate atmospheric oxidation under stable
conditions over a wide range of time periods and chemical scenarios. For example, a steady-state
NO level at ~1 ppb was created by the continuously mixed flow chamber operation for the study
of isoprene photooxidation chemistry (Liu et al., 2013). In this study, we present the
development and characterization of the NCAR Atmospheric Simulation Chamber, which is
operated in steady state continuous flow mode for simulating atmospheric daytime and nighttime
chemistry over chemical regimes not accessible in static chamber experiments. We focus on
establishing an 'intermediate NO' regime characterized by a constant steady-state NO level
ranging from tens of ppt to a few ppb in the chamber. This particular chemical regime is well
suited for the study of atmospheric behavior of $RO_2$ radicals, as they can survive up to minutes
and embrace various reaction possibilities as opposed to reaction with NO, $NO_3$, $HO_2$, and $RO_2$
as their dominant fate observed in most batch-mode chamber experiments. We employ the
'intermediate NO' regimes to reexamine the daytime and nocturnal chemistry of isoprene
through the measurements of two first-generation products, methacrolein (MACR) and methyl
vinyl ketone (MVK).
**2. Experimental**
**2.1 NCAR Atmospheric Simulation Chamber Facilities**
The NCAR Atmospheric Simulation Chamber consists of a 10 $m^3$ FEP Teflon (0.005" thick)
bag that is housed in a cubic enclosure with UV reflective surfaces and a bank of 128 wall-

mounted blacklight tubes (32W, Type F32T8/BL). To characterize photolytic conditions in the chamber, irradiance spectra were collected in the wavelength range of 180–600 nm at ~0.8 nm resolution by a custom-built spectroradiometer, as shown in Figure S1 in the Supplement (Petropavlovskikh et al., 2007). Photolysis frequencies were calculated based on the measured downwelling spectral actinic fluxes. The computed photolysis rate of $NO_2$ ($J_{NO_2} \sim 1.27 \times 10^{-3}$ s$^{-1}$) agrees within 3% with that measured by photolyzing 18.6 ppb $NO_2$ in the chamber and monitoring the NO production rate. The chamber is equipped with a standard set of measurements, including an integrated temperature and humidity probe (Model 50U, VAISALA, CO) and a Magnehelic differential pressure indicating transmitter (Model 605-11, Dwyer Instruments, IN). The chamber temperature is controlled at 295 K by the building's air conditioning system and increases to 305–306 K under maximum irradiation conditions. The relative humidity of the chamber air is below 10% under dry conditions (the remaining water vapor is generated from methane combustion during the air purification process) and can be varied in the range of ~10–50% by flowing a portion of the purified dry flushing air into the chamber through a temperature-controlled water reservoir. Typical temperature and relative humidity profiles across the chamber under maximum irradiation conditions are given in Figure S1 in the Supplement. The chamber internal pressure is maintained slightly above the ambient pressure to minimize the enclosure air contamination via penetration through the Teflon film.

Prior to each experiment, the chamber was flushed with purified dry air from an ultra high purity zero air generator (Model 737, Aadco Instruments, OH) for >12 h until ozone and $NO_x$ levels were below 1 ppb. During the operation of the steady state continuous flow mode, the chamber was constantly flushed with purified dry air at 40 L min$^{-1}$, which gives an average chamber residence time of 4.17 hours. The incoming and outgoing flows were balanced by a feedback control system that maintains a constant internal pressure of $1.2$–$4.9 \times 10^{-4}$ atm above the ambient. The chamber is actively mixed by the turbulence created by the 40 L min$^{-1}$ flushing air. The characteristic mixing time is defined as the time it takes for the measurement signal of a tracer compound (e.g., $CO_2$ and NO) to stabilize following a pulse injection. The average mixing time in the NCAR chamber was determined to be ~9 min, which is ~4% of the residence time. Under such conditions, the gas/particle-phase composition in the outflow can be assumed identical to that in the well-mixed core of the chamber.

To mimic daytime photochemistry in the continuous flow mode, steady-state OH mixing ratio was created by photolyzing hydrogen peroxide ($H_2O_2$) vapor that was continuously flowing into the chamber ($H_2O_2 + h\nu \rightarrow 2OH$, $J_{H_2O_2} \sim 3.93 \times 10^{-7}$ s$^{-1}$). Specifically, a 20 mL syringe (NORM-JECT, Henke-Sass Wolf, MA) held on a syringe pump (Model 100, kdScientific, MA) kept at ~4 °C was used to deliver $H_2O_2$ solution (1–30 wt%, Sigma Aldrich, MO) into a glass bulb that was gently warmed at ~32 °C. The liquid delivery rate was sufficiently slow (100–300 µL min$^{-1}$) that all the $H_2O_2$ vapor was released into the glass bulb through evaporation of a small droplet hanging on the needle tip. An air stream (5 L/min) swept the $H_2O_2$ vapor into the chamber, resulting in an $H_2O_2$ mixing ratio in the range of 600 ppb to 16.22 ppm in the injection flow as a function of the concentration of $H_2O_2$ aqueous solution used. A spreadsheet (Table S2) for calculating the inflow $H_2O_2$ mixing ratio using the above input method is provided in the Supplement. As $H_2O_2$-laden air was continuously entering the chamber, it took approximately three turnover times (~12.5 hr) for the desired $H_2O_2$ vapor mixing ratio to reach steady state in the chamber. The $H_2O_2$ vapor concentration in the chamber, though not measured, can be estimated from the steady-state OH mixing ratio derived from the observed exponential decay of

a given parent hydrocarbon. Constant NO injection flow was achieved by diluting the gas flow
from a concentrated NO cylinder (NO = 133.16 ppm, balance $N_2$) to a desired mixing ratio
(0.1–100 ppb) using a set of mass flow controllers (Tylan FC260 and FC262, Mykrolis Corp.,
MA). The lowest steady-state NO level that can be achieved in the chamber is around 30 ppt
(unpublished, NCAR). Note that for experiments performed in the absence of any VOC
precursor, $H_2O_2$ and NO were the only two species that were continuously input into the chamber
for the establishment of a combination of different photochemical conditions as denoted by the
$O_3$ and $NO_x$ measurements. For the isoprene photooxidation experiments, an isoprene standard
($C_5H_8$ = 531 ppm, balance $N_2$) was constantly injected into the chamber and diluted with the
flushing air to achieve an inflow concentration of ~20 ppb.
To mimic the nighttime chemistry in the continuous flow mode, steady-state $NO_3$ mixing
ratio was created by constantly flowing diluted $O_3$ and NO air into the chamber ($NO+O_3$
$\rightarrow NO_2+O_2$; $NO_2+O_3 \rightarrow NO_3+O_2$). The NO source can be replaced by $NO_2$, although the absolute
absence of NO does not necessarily represent the actual atmospheric conditions. $O_3$ was
produced by photolyzing $O_2$ in air at 185 nm using a mercury "Pen-Ray" lamp (UVP LLC, CA).
Ozone concentration in the injection flow can be controlled from 3.5 ppb to 457 ppb
automatically by adjusting the mercury lamp duty cycle. To study the $RO_2+HO_2$ pathway,
formaldehyde ($CH_2O$) was input into the chamber along with NO and $O_3$ to initiate $HO_2$
production ($NO_3+CH_2O+O_2 \rightarrow HNO_3+HO_2+CO$). Formaldehyde aqueous solution (37 wt%,
Sigma Aldrich, MO) was diluted with ultrapure water (Milli-Q, Merck Millipore, MA) to 0.2–1.0
wt% and continuously input into the chamber using the same method used for $H_2O_2$ input
described above. It is worth noting that the formaldehyde aqueous solution contains 10–15%
methanol as stabilizer to prevent polymerization. The presence of methanol in the chamber does
not significantly impact the nocturnal chemistry as it consumes OH and $NO_3$ radicals to generate
formaldehyde           and           additional           $HO_2$           ($CH_3OH+NO_3 \rightarrow HNO_3+CH_2O+HO_2$,
$CH_3OH+OH \rightarrow H_2O+CH_2O+HO_2$) (Atkinson et al., 2006). The use of formaldehyde as an $HO_2$
source mimics closely the atmospheric nighttime conditions in forest environments (Schwantes
et al., 2015). To study the $NO_3$-initiated oxidation of isoprene, an injection flow of diluted
isoprene (~10 ppb) was achieved using the procedure described above.
**2.2 Analytical measurements**
A suite of instruments was used to monitor gas-phase concentrations in the chamber
outflow. $O_3$ was monitored by absorption spectroscopy with 0.5 ppb detection limit (Model 49,
Thermo Scientific, CO). The $O_3$ monitor was calibrated using an Ozone Primary Standard in the
range of 0 to 200 ppb (Model 49i-PS, Thermo Scientific, CO). The $O_3$ monitor used for chamber
experiments was periodically checked with the primary standard and was shown to be stable over
long periods of time (less than 1 ppb drift in over 1 year). NO was monitored by
chemiluminescence with 0.5 ppb detection limit (Model CLD 88Y, Eco Physics, MI). Zero-point
and span calibrations of the $NO_x$ monitor were performed prior to each experiment by supplying
the instrument with pure $N_2$ gas and diluted NO, respectively. Multi-point calibration was
performed on a weekly basis and has shown a good stability and linearity in the NO
measurement ranging from 1 ppb to 200 ppb. $NO_x$ ($NO+NO_2$) measurements were performed
using a photolytic converter that selectively converts $NO_2$ to NO upstream of the photo-
multiplier tube in the CLD 88Y NO monitor.  This converter uses two opposing arrays of UV
LEDs shining into a cylindrical quartz mixing tube to achieve approximately 50% conversion of
$NO_2$ to NO per second.  The total efficiency for the equipment described here is approximately
70% as determined by measuring calibrated $NO_2$ standards. The sample path always includes
the photolytic converter, and typical experiments cycle the power for the LED lights to switch
between measuring NO (lights off) and $NO_x$ (lights on). $NO_2$ concentrations were then
determined by subtracting the NO from the adjacent $NO_x$ measurements.
A customized Proton Transfer Reaction Quadrupole Mass Spectrometer (PTR-Q-MS) was
used to measure volatile organic compounds including isoprene ($C_5H_8$), methacrolein (MACR,
$C_4H_6O$), and methyl vinyl ketone (MVK, $C_4H_6O$). The instrument was operated at 2.3 mbar drift
pressure and 560 V drift voltage. Measurements reported here were obtained at a sampling rate
of 10 Hz. In positive-mode operation, a given analyte [M] undergoes proton transfer reaction,
producing an ion of the form $[M+H]^+$, that is, isoprene is detected as ion $C_5H_9^+$ ($m/z$ 69) and
MACR and MVK are both detected as ion $C_4H_7O^+$ ($m/z$ 71). The instrument background was
collected by sampling the chamber air for at least 30 min prior to each experiment. Measured ion
intensities for isoprene ($C_5H_9^+$, $m/z$ 69) and MACR and MVK ($C_4H_7O^+$, $m/z$ 71) were calculated
as the signal of each ion (counts per seconds) normalized to the total ion signal of $H_3O^+$. The
instrument sensitivities towards isoprene, MACR and MVK were calibrated with a mixture of
diluted gas standards. The instrument sensitivity towards MACR is identical to that of MVK, and
as a result, the sum of MACR and MVK concentration in the sampling air can be calculated by
applying one calibration factor to the measured $C_4H_7O^+$ ($m/z$ 71) signal intensity. Since artifacts
in the measured $C_4H_7O^+$ signal can be produced through thermal decomposition of isoprene
oxidation products, such as the peroxides, nitrates, and epoxides, on contact with hot metal
surface (Liu et al., 2013; Nguyen et al., 2014b; Rivera-Rios et al., 2014), a cold-trap system was
used to avoid bias in the interpretation of the PTR-MS data. Specifically, a 1 m section of Teflon
tubing was submerged in a low temperature ethanol bath ($-40\pm2$ °C) that could trap oxidized
products less volatile than the authentic MACR and MVK standards after steady state was
established in the chamber. The quantification of the sum of MACR and MVK was then based
on the PTR-MS measured $C_4H_7O^+$ ($m/z$ 71) signal downstream of the cold-trap.

## 3. Kinetic Modeling

Reaction kinetics and mechanisms for the gas-phase photochemistry were extracted from the
Master Chemical Mechanism (MCMv3.3.1, accessible at http://mcm.leeds.ac.uk/MCM/). The
inorganic reaction scheme includes 21 species and 48 reactions; and the isoprene oxidation
system includes 611 species and 1974 reactions. The kinetic schemes were implemented in
Matlab (Mathworks) to simulate the temporal profile of a given compound $i$ in the chamber
operated in the steady state continuous flow mode:

$$\frac{dC_i}{dt} \cdot \tau = C_{i,0} + P_i - C_i - \sum k_i \cdot \tau \cdot C_i \qquad \text{(Eq1)}$$

where $C_i$ (molec $cm^{-3}$) is the gas-phase concentration of compound $i$ in the well-mixed core of
the chamber; $C_{i,0}$ (molec $cm^{-3}$) is the initial gas-phase concentration of compound $i$ in the
injection flow; $k_i$ ($s^{-1}$) is the pseudo-$1^{st}$-order rate coefficient for a chemical reaction that
consumes compound $i$; $\tau$ (s) is the chamber mean residence time and can be calculated as the
total chamber volume divided by the incoming/outgoing flow rate; and $P_i$ (molec $cm^{-3}$) is the
increment in the concentration of compound $i$ through chemical production during one residence
time. Note that two terms are neglected in Equation (1), i.e., organic vapor condensation onto
particles and deposition on the chamber wall. This is a reasonable simplification here owing to
the relatively high volatility ($\geq 10^{-1}$ atm) of compounds studied (Zhang et al., 2015b; Krechmer
et al., 2016; Huang et al., 2018). Incorporation of these two terms into Equation (1) is feasible
given the vapor pressure of compound $i$, suspended particle size distribution, gas-particle and
gas-wall partitioning coefficient, accommodation coefficients of compound $i$ on particles and
walls, and the effective absorbing organic masses on the wall (Zhang et al., 2014a; Zhang et al.,
2015b; Huang et al., 2016; McVay et al., 2016; Nah et al., 2016).

252       Model simulations used for comparison with chamber measurements were initialized using
experimental conditions summarized in Table S1 in the Supplement. Model input parameters for
all simulations include temperature (295 at dark and 306 K under irradiation), local pressure
($8.6 \times 10^4$ Pa), relative humidity (8% at dark and 5% under irradiation), light intensity
($J_{NO_2} = 1.27 \times 10^{-3}$ s$^{-1}$ under irradiation and 0 at dark), chamber mean residence time (4.17 h), and
input mixing ratios of $H_2O_2$ (0.11−16.2 ppm for photolytic experiments), NO (0.1−100 ppb for
photolytic experiments and 10−20 ppb for dark experiments), $O_3$ (22−225 ppb for dark
experiments), HCHO (0−600 ppb for dark experiments), and isoprene (19.9 ppb for photolytic
experiments and 10.2 ppb for dark experiments). The model was propagated numerically for 25 h
duration for each experiment.

## 4. Results and Discussions

### 4.1 Optimal operating conditions for daytime photochemistry

264       Figure 1 shows the model predicted steady-state mixing ratios of OH, $HO_2$, $NO_3$, NO, $NO_2$,
and $O_3$ in the chamber after 20 hours of photochemical reactions as a function of the $H_2O_2$ and
NO concentrations in the continuous injection flow. Six blank chamber experiments were
compared with simulations. In general, the model captures the evolution patterns of $NO_x$ and $O_3$
well. The predicted mixing ratios of NO, $NO_2$, and $O_3$ agree within 69%, 11%, and 33%,
respectively, with the measurements (see Table S1 and Figure S2 in the Supplement). The
relatively large NO uncertainties originate from the measurements that were performed close to
the instrument detection limit (0.5 ppb).

272       Simulated steady-state mixing ratios of OH radicals ([OH]$_{ss}$) range from $\sim 5 \times 10^5$ to $\sim 4 \times 10^6$
molec cm$^{-3}$, which over $\sim 4$ hours chamber residence time would be roughly equivalent to $\sim 1$ h to
$\sim 8$ h of atmospheric OH exposure ($1 \times 10^6$ molec cm$^{-3}$). As expected, [OH]$_{ss}$ increases with
increasing NO influxes due to the enhanced $NO_x/O_3$ cycling but decreases with increasing $H_2O_2$
influxes owing to the overwhelming reaction $OH + H_2O_2 \rightarrow H_2O + HO_2$. As a consequence, the
steady-state mixing ratios of $HO_2$ radicals ([HO$_2$]$_{ss}$) reach up to $\sim 7 \times 10^9$ molec cm$^{-3}$ when 16.2
ppm $H_2O_2$ is continuously injected into the chamber. If 110 ppb $H_2O_2$ is used instead, the
resulting [HO$_2$]$_{ss}$ levels fall close to the ambient range ($\sim 10^8$ molec cm$^{-3}$).

280       Simulated steady-state NO mixing ratios in the chamber range from $\sim 2$ ppt to $\sim 0.9$ ppb from
combinations of 0.1−20 ppb NO and 0.11−16.22 ppm $H_2O_2$ in the injection flow. The ratio of
inflow NO concentration to the steady-state NO concentration in the chamber ranges from 5 to
93, demonstrating the importance of chemical removal in controlling the overall steady-state NO
levels. $O_3$ accumulation is an inevitable consequence under photolytic conditions and, for
example, the presence of 10 ppb $O_3$ leads to the chemical removal term ($k_{O_3+NO} \cdot [O_3] \cdot \tau$) in
Equation (1) that reduces the steady-state NO concentration by a factor of 60. It is worth noting
that under all simulated conditions in the continuous flow mode, $O_3$ ($\sim 1$−126 ppb) coexists with
NO (~0.002–0.9 ppb). This particular chemical scenario, which is impossible to achieve in
batch-mode reactors due to prompt conversion of NO to $NO_2$, could then be used to mimic
ambient ozonolysis chemistry, for example, in forest regions that frequently encounter polluted
air masses from nearby urban areas. The steady-state mixing ratios of $NO_2$ ($[NO_2]_{ss}$) exhibit a
strong linear correlation with NO influxes. The use of less than 20 ppb NO in the injection flow
results in a few to tens of ppb $[NO_2]_{ss}$ that is higher than the range typically found in the ambient.
The potential 'quenching' effect of $NO_2$ on $RO_2$ radicals through reversible termolecular
reactions is discussed shortly.
In the so-called 'high-$NO_x$' chamber experiments, the $NO_3$ radical is an unavoidable side-
product when black lights are used as a representative of the solar radiation in mimicking the
daytime photochemistry in the troposphere. The photolysis of $NO_3$, although its primary sink in
the atmosphere, proceeds rather slowly ($J_{NO_3}\sim 1.8\times10^{-3}$ $s^{-1}$) under the present chamber
photolytic conditions, thereby leading to a significant accumulation of $NO_3$ radicals
($7.9\times10^4$–$2.8\times10^8$ molec $cm^{-3}$) at steady state. The simulated $NO_3$/OH ratio dictates the extent to
which the $NO_3$ (nighttime) chemistry competes with the OH-initiated (daytime) photochemistry.
For compounds that are highly reactive towards $NO_3$ such as isoprene, $NO_3$-initiated oxidation
accounts for up to ~60% of the overall isoprene degradation pathways at the highest $NO_3$/OH
ratio (~255) simulated. Low concentrations of NO (< 20 ppb) and $H_2O_2$ (< 2 ppm) in the
injection flow are therefore necessary to limit the interferences of $NO_3$-initiated chemistry.
Again, taking isoprene as an example, the $NO_3$ oxidation pathway contributes less than 0.1% of
the overall isoprene degradation kinetics at the lowest $NO_3$/OH ratio (~0.13) simulated here.
Also given in Figure 1 is the calculated lifetime ($\tau_{RO_2}$) of an $RO_2$ radical with respect to
reactions with NO and $HO_2$ at 306 K. In most batch-mode chamber experiments, $\tau_{RO_2}$ of only
several seconds or less can be achieved, due to the presence of tens to hundreds of ppb levels of
NO and $HO_2$. Here $\tau_{RO_2}$ could extend to 60 s or even longer with the continuous input of low
mixing ratios of $H_2O_2$ (≤ 110 ppb) and NO (≤ 0.2 ppb). Note that the presence of tens of ppb
$NO_2$ in the chamber might impose a 'quenching' effect on the steady state $RO_2$ level through
rapid reversible reactions ($RO_2$+$NO_2$+M↔$RO_2NO_2$+M). We evaluate this potential 'quenching
effect' using ethylperoxy radical ($C_2H_5O_2$) generated from OH-oxidation of ethane as an
example. Simulations shown in Figure S3 in the Supplement reveal that incorporation of the
$C_2H_5O_2$+$NO_2$+M↔$C_2H_5O_2NO_2$+M reaction into the mechanism in the presence of ~1–80 ppb
$NO_2$ does not notably change the behavior of $C_2H_5O_2$ radical. One exception is the peroxyacyl
radical, which combines with $NO_2$ yielding peroxyacyl nitrate. For example, under 0.1–16 ppb
$[NO_2]_{ss}$ as displayed in Figure 1, we calculate that the time needed for peroxyacetyl radical
($CH_3C(O)O_2$)      to      reach      equilibrium      with      peroxyacetyl      nitrate
($CH_3C(O)O_2$+$NO_2$+M↔$CH_3C(O)O_2NO_2$+M) ranges from ~1 to ~10 s, suggesting that the
lifetime of peroxyacyl radicals is ultimately controlled by $NO_2$ instead of NO/$HO_2$ in the reaction
system, and consequently, peroxyacyl radicals are not expected to be long-lived in the current
chamber configuration.
We further compare the photochemical oxidation environment created here with the
'intermediate-NO' conditions achieved by other chambers that employed the experimental
approaches introduced earlier. In terms of the oxidizing power, all approaches are capable of
maintaining an atmospheric relevant OH level (~$10^6$ molec $cm^{-3}$), expect the 'slow chemistry'
scenario that limits the photolysis rate of the OH precursor and results in an average OH mixing
ratio of ~$10^5$ molec cm$^{-3}$ (Crounse et al., 2012; Teng et al., 2017). At comparable OH levels, the
overall atmospheric OH exposure achieved in the flow tube reactor is rather limited due to the
short residence time (e.g., ~80 s in the PAM reactor). In terms of the NO$_x$ level, precisely
controlled steady-state NO concentration can be achieved for an indefinite time period by
operating chambers in the continuously mixed flow mode. However, NO$_2$ accumulates during
the continuous oxidation process and the resulting NO$_2$/NO ratio can be as much as an order of
magnitude higher than that achieved in the static outdoor chambers.
**4.2 Application to OH-initiated oxidation of isoprene**
Methacrolein (MACR) and methyl vinyl ketone (MVK) are major first-generation products
from the OH-initiated oxidation of isoprene in the presence of NO (Wennberg et al., 2018). They
are produced from the decomposition of β-ISOPO alkoxy radicals that are primarily formed from
the reaction of β-ISOPOO peroxy radicals (β-1-OH-2-OO and β-4-OH-3-OO) with NO, see
mechanisms displayed in Figure 2 (A). Reactions of β-ISOPOO peroxy radicals with HO$_2$ and
RO$_2$ also partially yield β-ISOPO alkoxy radicals that ultimately lead to MACR and MVK,
although these pathways are considered to be minor in the presence of hundreds to thousands of
ppt NO in the atmosphere. The molar yields determined from previous studies range from
30–35% for MVK and 20–25% for MACR under high-NO conditions (NO > 60 ppb) (Tuazon
and Atkinson, 1990; Paulson and Seinfeld, 1992; Miyoshi et al., 1994; Ruppert and Becker,
2000; Sprengnether et al., 2002; Galloway et al., 2011; Liu et al., 2013). It has been recently
shown that the six hydroxyl peroxy radicals (ISOPOO) produced from the initial OH addition to
the double bonds of isoprene undergo rapid interconversion by removal/addition of O$_2$ that
competes with bimolecular reactions under atmospherically relevant NO levels (Peeters et al.,
2014; Teng et al., 2017). As a result, the distribution of ISOPOO radical isomers and their
subsequent reaction products varies with their lifetimes with respect to bimolecular reactions. In
the presence of hundreds of ppb NO as done by most previous experimental studies, the reaction
of ISOPOO radicals with NO dominates over their interconversion, and the production of β-
ISOPOO peroxy radical is less favored, leading the experiments to underestimate the MACR and
MVK yields typically obtained in ambient conditions. Measurements by Karl et al. (2006) and
Liu et al. (2013) conducted at NO concentrations comparable to the moderately polluted urban
environment (~ 0.2 ppb in Karl et al. and ~1 ppb in Liu et al.) found higher MACR (~27% in
Karl et al. and ~31.8% in Liu et al.) and MVK (~41% in Karl et al. and ~44.5% in Liu et al.)
yields than other studies.
Here we perform a steady-state continuous-mode experiment to measure the production of
MACR and MVK from the OH-initiated oxidation of isoprene in the presence of ~0.45 ppb NO.
Figure 3 shows the observed and simulated temporal profiles of NO$_x$, O$_3$, C$_5$H$_8$, and C$_4$H$_6$O over
24 hours photooxidation of isoprene. In this experiment, C$_5$H$_8$, H$_2$O$_2$, and NO were continuously
fed into the chamber, with constant inflow concentrations of 19.9 ppb, 600 ppb, and 19 ppb,
respectively. An outgoing flow at 40 L min$^{-1}$ continuously withdrew air from the chamber to
balance the pressure. After approaching steady state, the sampling tube was submerged into an
ethanol low temperature bath (–40±2 °C) to trap oxidized products that would otherwise undergo
thermal decomposition introducing interferences in the C$_4$H$_7$O$^+$ (*m/z* 71) signal. The measured
concentrations of C$_5$H$_8$ and C$_4$H$_6$O upon cold-trapping agree within 2.8% and 4.6% uncertainties
with the model simulations, see Fig.3 (C) and (D).

375   To calculate the total molar yield ($Y_{C_4H_6O}$) of MACR and MVK, two reactions are
376 considered:

377        $$C_5H_8 + OH \xrightarrow{k_1} Y_{C_4H_6O} \cdot C_4H_6O + products \tag{R1}$$

378          $$C_4H_6O + OH \xrightarrow{k_2} products \tag{R3}$$

379 where $k_1$ is the rate constant for OH reaction with isoprene, and $k_2$ is taken as the average of rate
380 constants for OH reactions with MACR and MVK. Uncertainties associated with the
381 simplification of $k_2$ in calculating the MACR and MVK yields will be discussed shortly. Note
382 that the ozonolysis and $NO_3$-initiated oxidation in total account for less than 6% of isoprene
383 degradation pathway under current experimental conditions and are neglected in the calculation.
384 The ozonolysis and photolysis in total account for ~6% of the $C_4H_6O$ degradation pathway and
385 are neglected here as well.

386   In the continuous-mode operation, two mass conservation equations are satisfied at steady
387 state:

388      $$\frac{d[C_5H_8]_{ss}}{dt} = [C_5H_8]_0/\tau - [C_5H_8]_{ss}/\tau - k_1 \cdot [OH]_{ss} \cdot [C_5H_8]_{ss} = 0 \tag{Eq2}$$

389 $$\frac{d[C_4H_6O]_{ss}}{dt} = Y_{C_4H_6O} \cdot k_1 \cdot [OH]_{ss} \cdot [C_5H_8]_{ss} - k_2 \cdot [OH]_{ss} \cdot [C_4H_6O]_{ss} - [C_4H_6O]_{ss}/\tau = 0 \tag{Eq3}$$

390 where $[C_5H_8]_{ss}$ and $[C_4H_6O]_{ss}$ are the PTRMS measured steady-state concentrations of isoprene
391 and the sum of MACR and MVK when using the cold trap, respectively, $[C_5H_8]_0$ is the initial
392 concentration of isoprene, and $\tau$ is the chamber mean residence time and can be calculated as the
393 total chamber volume divided by the incoming/outgoing flow rate. The steady state OH radical
394 concentration ($[OH]_{ss}$) can be derived by solving Equation (2). The calculated $[OH]_{ss}$ ($3.13 \times 10^6$
395 molec $cm^{-3}$) is 12% higher than the model prediction ($2.74 \times 10^6$ molec $cm^{-3}$). The molar yield of
396 the sum of MACR and MVK from isoprene OH oxidation pathway in the presence of ~0.45 ppb
397 NO is thus given by Equation (4) and calculated as 76.7±5.8%:

398      $$Y_{C_4H_6O} = \frac{[C_4H_6O]_{ss} + k_2 \cdot [OH]_{ss} \cdot \tau \cdot [C_4H_6O]_{ss}}{k_1 \cdot [OH]_{ss} \cdot \tau \cdot [C_5H_8]_{ss}} \times f_{\beta\text{-ISOPOO+NO}} \tag{Eq4}$$

399 Here a 5.8% uncertainty originates from the assumption that MACR+OH and MVK+OH
400 proceed with equal reaction rate, although the rate constant for MVK+OH is ~31% lower than
401 that of MACR+OH. Another potential uncertainty relates to the accuracy of the simulated steady
402 state $HO_2$ and $RO_2$ concentrations and the contribution of β-ISOPOO+$HO_2$ and β-ISOPOO+$RO_2$
403 reaction pathways to the overall β-ISOPOO fate. The fraction of β-ISOPOO radicals that reacts
404 with NO ($f_{\beta\text{-ISOPOO+NO}}$) was predicted as 0.88, which was used here to scale the final
405 MACR+MVK yield, see Eq(4).

406   In summary, the measured yield of the sum of MACR and MVK in this study is close to that
407 reported by Karl et al. (2006) and Liu et al. (2013), but ~27–52% higher than the majority of
408 previous measurements performed under high-NO conditions (NO > 60 ppb). This is consistent
409 with the dynamic nature of the six ISOPOO radical isomers that undergo rapid interconversion
410 by addition/removal of $O_2$. In the presence of ~0.45 ppb NO as performed in this study, lifetimes
411 of the β-1-OH-2-OO peroxy radical with respect to reaction with NO and loss of $O_2$ are

estimated as 9.4 s and 0.2 s, respectively, implying that the rapid interconversion between β-
ISOPOO and δ-ISOPOO radicals essentially governs their distribution, and under such
conditions, the production of thermodynamically more stable β-ISOPOO isomers is favored,
leading to higher yields of MACR and MVK. Here the reported MACR and MVK yield from
isoprene OH oxidation in the presence of ~0.45 ppb NO represents an illustration of chamber
operation at steady state continuous flow mode for the establishment of certain experimental
conditions that are not easily accessible from traditional batch-mode chamber experiments. A
complete measurement of first-generation oxidation products from isoprene OH reaction under a
wide range of NO levels (ISOPOO bimolecular lifetimes) will be forthcoming in a future
publication.
**4.3 Optimal operating conditions for nighttime chemistry**
Figure 4 shows the model simulated steady-state mixing ratios of $HO_2$, $NO_3$, NO, $NO_2$, and
$O_3$ after 16 hours of dark reactions in the chamber as a function of the HCHO concentration and
$O_3$/NO ratio in the continuous chamber inflow. Blank experiments were compared with
simulations in five cases (see Table S1 in the Supplement). The model captures the evolution
patterns of $NO_x$ and $O_3$ well. The observed mixing ratios of $NO_2$ and $O_3$ agree with the
simulations to within 11% and 6%, respectively (Figure S4 in the Supplement).
Compared with the photochemical reaction schemes discussed earlier, the nocturnal
chemistry is rather straightforward; that is, the inflow $O_3$/NO ratio governs the steady-state
concentrations of $NO_3$, $NO_x$, and $O_3$, while the inflow HCHO concentration ultimately controls
the steady-state $HO_2$ level. Increasing the $O_3$/NO ratio from 1 to 9 in the continuous inflow leads
to increased $NO_3$ from $2.4 \times 10^5$ to $1.1 \times 10^9$ molec cm$^{-3}$, but decreased NO from 1.8 ppb to 20 ppt
and decreased $NO_2$ from 18 to 7 ppb. At a fixed inflow $O_3$/NO ratio, doubling the NO and $O_3$
concentrations leads to elevated $NO_3$, $NO_x$, and $O_3$ by a factor of 2.0–3.2, 1.5–2.0, and 1.4–2.0,
respectively. The use of HCHO as an effective dark $HO_2$ source does not significantly impact the
steady-state mixing ratios of $NO_x$ and $O_3$, but slightly weakens the $NO_3$ production.
The calculated $RO_2$ lifetime ($\tau_{RO_2}$) with respect to reactions with NO, $NO_3$, and $HO_2$ at 295
K ranges from 3 to 225 s. The highest $\tau_{RO_2}$ was achieved in the absence of any HCHO source
and corresponds to a chemical regime that can be employed to study the intramolecular
isomerization (autoxidation) pathway of $RO_2$ radicals, if any. Adding a continuous flow of
HCHO to the system leads to the production of $10^7$–$10^9$ molec cm$^{-3}$ $HO_2$ radicals that then
constitute a significant sink of $RO_2$ radicals and represents prevailing forest environments during
nighttime.
**4.4 Application to $NO_3$-initiated oxidation of isoprene**
$NO_3$-initiated oxidation of isoprene proceeds by the $NO_3$ addition to the carbon double
bonds followed by $O_2$ addition, yielding six distinct nitrooxy peroxy radicals (INOO), including
two isomers (β-INOO) with $O_2$ added on the β-carbon to the nitrate group (see Figure 2B for
schematic illustration). The β-INOO radicals react further with $NO_3$, $HO_2$, NO, and $RO_2$,
producing nitrooxy alkoxy radicals (β-INO) with molar yields of 1.00, 0.53, 0.97, and 0.40,
respectively (Wennberg et al., 2018). The further decomposition of β-INO radicals produces
MACR and MVK, together with HCHO and $NO_2$. Depending on the actual fate of β-INOO
radicals, the yields of β-INO radicals can then vary from 0.4 to 1.0, resulting in a distinct
distribution of final oxidation products. It is thus important to specify the ultimate fate of INOO
radicals during quantification of oxidation products from isoprene reaction with $NO_3$. As an
illustration, we performed one continuous mode experiment that targets on controlling the
steady-state fate of INOO radicals to be their reaction with NO and $HO_2$ (46% and 38% INOO
radicals are predicted to react with NO and $HO_2$, respectively, as shown in Fig.5). Note that by
adjusting the concentrations and fractions of inflow reactants ($O_3$, NO, HCHO, and $C_5H_8$),
different chemical fates and lifetimes of INOO radicals can be achieved.
Figure 5 shows the observed and predicted temporal profiles of $NO_x$, $O_3$, $C_5H_8$, and $C_4H_6O$
over 25 hours of isoprene oxidation by $NO_3$, with continuous input of 10.2 ppb $C_5H_8$, 205 ppb
$O_3$, and 59 ppb NO into the chamber and a balancing outgoing flow at 40 L min$^{-1}$ carrying well-
mixed reactants and products. It took >16 hours to reach steady state for all the species
displayed. In general, the model captures the trends of $O_3$ and NO well, while underpredicting
the steady state $NO_2$ by ~26%. After ~18 hours of dark reaction, the PTR-MS sampling tubing
was submerged into an ethanol cold bath (– 40±2 °C) to trap artifacts in the PTR-MS measured
$C_4H_7O^+$ (*m/z* 71) signal. The simulated steady-state concentration of isoprene agrees within 9%
with the measurements. The derived concentration of the sum of MACR and MVK from the
measured $C_4H_7O^+$ ion intensity upon cold-trapping is ~1.1 ppb, which is ~129% higher than the
model predictions (~0.48 ppb). This disagreement can be attributed, to a large extent, to the
oversimplified representation of the six different INOO radicals as one δ-INOO isomer in the
MCMv3.3.1 mechanism. As a result, the production of β-INOO radical, the important precursor
of MACR and MVK, from $NO_3$-initiated oxidation of isoprene is suppressed in the simulations.
The measured molar yield of the sum of MACR and MVK is 36.3±12.1%, with uncertainties
arising from the fact that 10.5% isoprene is predicted to react with OH as an additional source of
MACR and MVK. Using this value, the fraction of β-INOO over the sum of nitrooxy peroxy
radicals is estimated as 48.6±16.2%, which is close to that (~46.3%) reported by Schwantes et al.
(2015), although the estimated bimolecular lifetime of INOO radical in that study (~30 s) is
lower than that predicted in the present work (~50 s). As discussed above, the hydroxyl peroxy
radicals produced from OH-oxidation of isoprene could undergo rapid interconversion through
addition/removal of $O_2$ at atmospherically relevant lifetimes. This interconversion significantly
impacts the subsequent chemistry of individual ISOPOO radical isomers in terms of reaction
rates and product distributions. It is likely that the INOO radicals follow similar interconversion
due to the small R-OO bond dissociation energy, although no experimental evidence exists. A
full examination of the INOO chemistry, i.e., their kinetic and thermodynamic properties as well
as their chemical fate at different lifetimes, will be the focus of future studies using this
continuous flow chamber operation method.
**5. Conclusions**
We report here the development and characterization of the NCAR Atmospheric Simulation
Chamber operated at steady state continuous flow mode for simulating daytime and nocturnal
chemistry under atmospherically relevant NO levels. The chamber is designed to achieve a well-
controlled steady-state environment by continuous inflow of reactants and continuous
withdrawal of reactor contents. We use a combination of kinetic modeling and chamber
experiments to characterize the 'intermediate-NO' chemical regime (tens of ppt to a few ppb)
that can be achieved by precisely controlling the inlet reactant concentrations and the
mixing/residence timescales of the chamber.
To mimic daytime photochemistry, continuous input of $H_2O_2$ and NO gases is required,
resulting in steady state OH mixing ratios of $10^5–10^6$ molec cm$^{-3}$ under irradiation. Under such
conditions, the lifetime of a peroxy radical with respect to reaction with NO and $HO_2$ can be
extended to 60 s or even longer, thus providing a unique environment to study all reaction
possibilities of $RO_2$ radicals including the intramolecular isomerization (autoxidation) pathway.
When studying OH-initiated chemistry, care needs to be taken to avoid a range of experimental
conditions (e.g., inflow $H_2O_2 > 2$ ppm and NO > 20 ppb) where $NO_3$-oxidation might account for
a large fraction of the overall degradation pathway of certain parent hydrocarbons such as
alkenes.
To mimic nighttime chemistry, continuous input of NO (or $NO_2$) and $O_3$ is needed to
produce steady state $NO_3$ radicals in the range of $10^6–10^9$ molec cm$^{-3}$ in the dark. Under such
conditions, an $RO_2$ radical can live up to 4 min prior to finding a bimolecular reaction partner
(e.g., NO, $NO_3$, and $HO_2$), which were the dominant fates of $RO_2$ radicals in most batch-mode
chamber experiments. Again, the long lifetime of $RO_2$ radicals achieved by the steady state
continuous mode operation opens an avenue for close examination of $RO_2$ unimolecular
(isomerization) pathways in nocturnal environments.
In simulating both daytime and nighttime chemistry with continuous flow operation method,
$O_3$ accumulation is unavoidable. The extent to which ozonolysis interferes with OH- or $NO_3$-
initiated oxidation chemistry depends on the steady state $O_3$ concentration achieved in the
chamber and its reactivity towards various parent VOCs. Taking isoprene as an example,
ozonolysis accounts for <1% and <0.1% of the overall isoprene degradation kinetics,
respectively, under established steady-state photolytic and dark conditions described above.
In atmospheric chemistry, the terms 'zero-NO' versus 'high-NO' have been widely used to
classify photooxidation conditions and delineate the gas-phase fate of the peroxy radicals ($RO_2$)
generated from VOCs oxidation (Zhang et al., 2010; He et al., 2011; Cappa et al., 2013; Zhang
and Seinfeld, 2013; Loza et al., 2014; Nguyen et al., 2014a; Schilling Fahnestock et al., 2014;
Zhang et al., 2014b; Krechmer et al., 2015; Zhang et al., 2015a; Gordon et al., 2016; Riva et al.,
2016; Thomas et al., 2016; Schwantes et al., 2017a; Schwantes et al., 2017b). In the so-called
'high-NO' regime, reaction with NO dominates the fate of $RO_2$ radicals, whereas in the 'zero-
NO' regime, the $RO_2$ radicals primarily undergo reaction with $HO_2$ and, perhaps to a much lesser
degree, self/cross-combination. The importance of the 'intermediate-NO' regime lies in the fact
that at sub-ppb levels of NO, the $RO_2$+NO and $RO_2$+$HO_2$ reactions are expected to co-exist and
the $RO_2$ radical could survive up to several minutes before encountering a partner (NO/$HO_2$) for
bimolecular reactions. Under such conditions, the $RO_2$ radical isomers may undergo
interconversion by addition/removal of $O_2$ and intramolecular isomerization (autoxidation)
through H-shift. Here we use isoprene as an illustrative VOC to explore the fate of $RO_2$ radicals
under sub-ppb NO. Future work will focus on detailed characterization of oxidation products
from isoprene day- and nighttime chemistry with particular attention given to the controlled $RO_2$
fates and lifetimes.

**Data Availability**

Data presented in this manuscript are available upon request to the corresponding author.

**Competing interests**

The authors declare that they have no conflict of interest.

**Acknowledgement**

The National Center for Atmospheric Research is operated by the University Corporation for Atmospheric Research, under the sponsorship of the National Science Foundation.

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

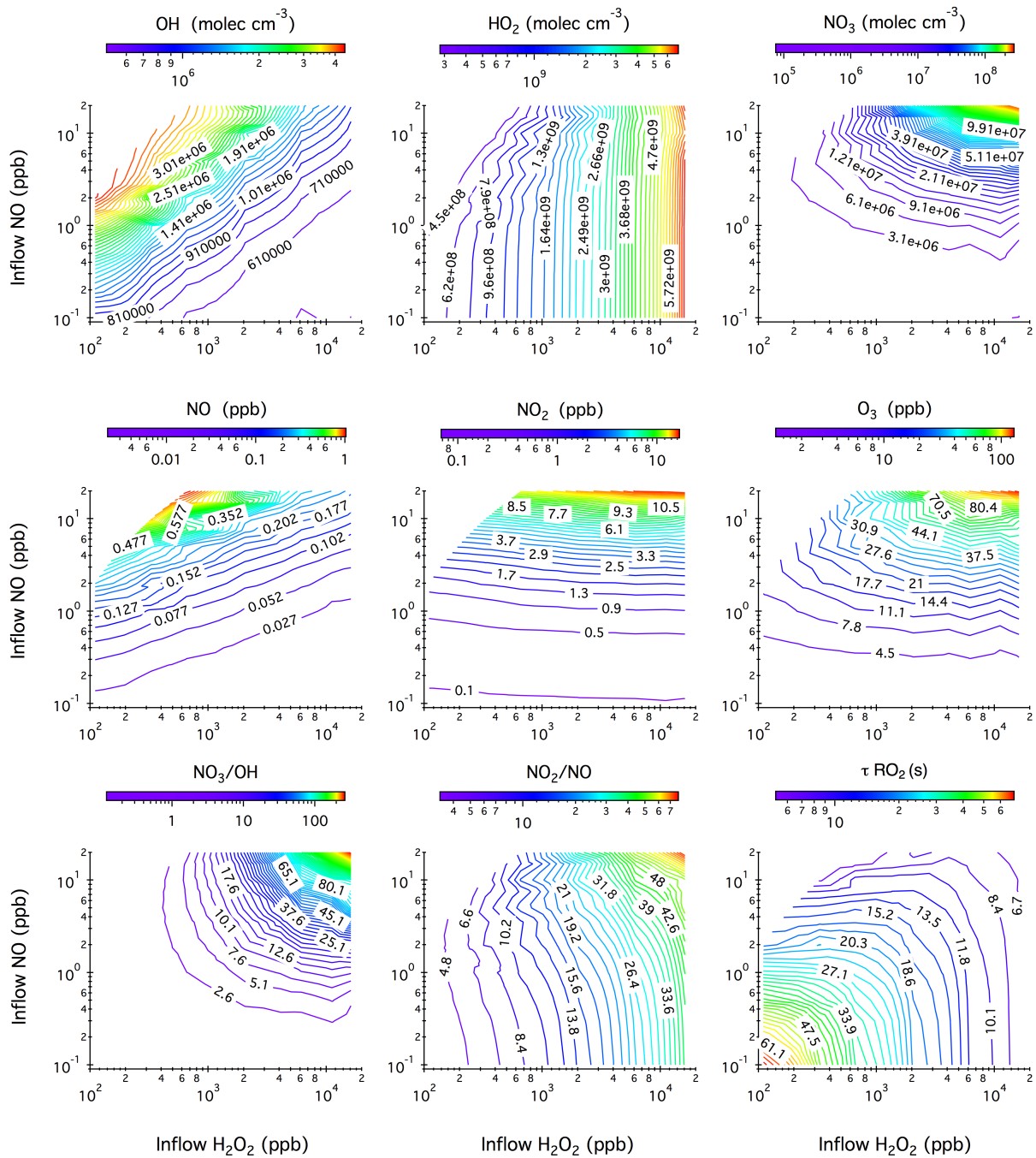

Figure 1. Contour plots showing the model predicted steady-state mixing ratios of OH, HO$_2$, NO$_3$, NO, NO$_2$, and O$_3$ after 20 hours of photochemical reactions in the chamber as a function of the concentrations of H$_2$O$_2$ and NO in the continuous injection flow. Also given here are the simulated NO$_3$ to OH ratio, NO$_2$ to NO ratio, and the lifetime of an RO$_2$ radical ($\tau_{RO_2}$) with respect to reactions with NO and HO$_2$. Note that the ripples on the contour lines originate from the limited simulation datasets that are used to generate iso-response values.

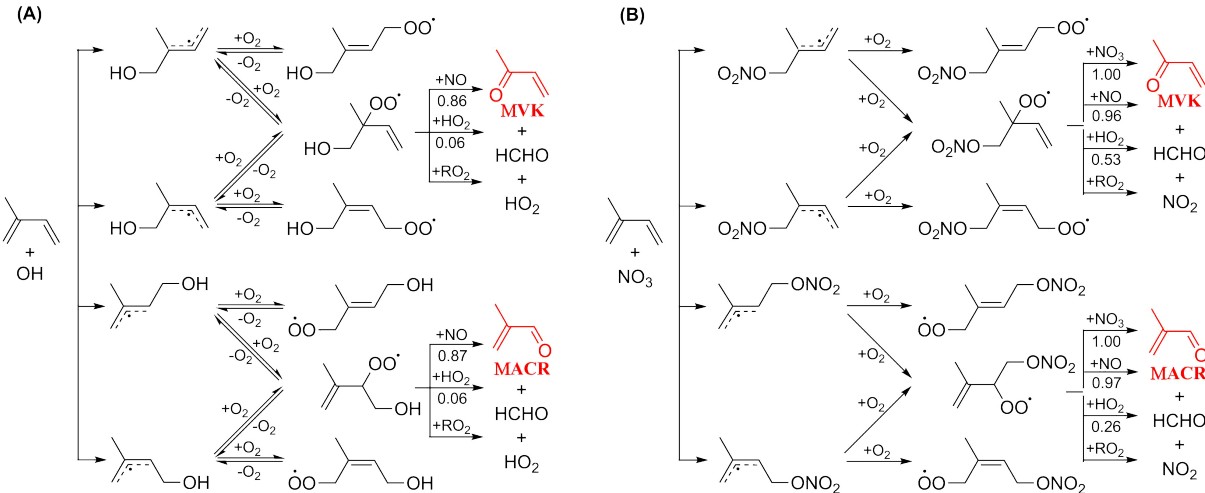

Figure 2. Representative mechanism for (A) OH- and (B) NO$_3$-initiated oxidation of isoprene that leads to the formation of MACR and MVK.

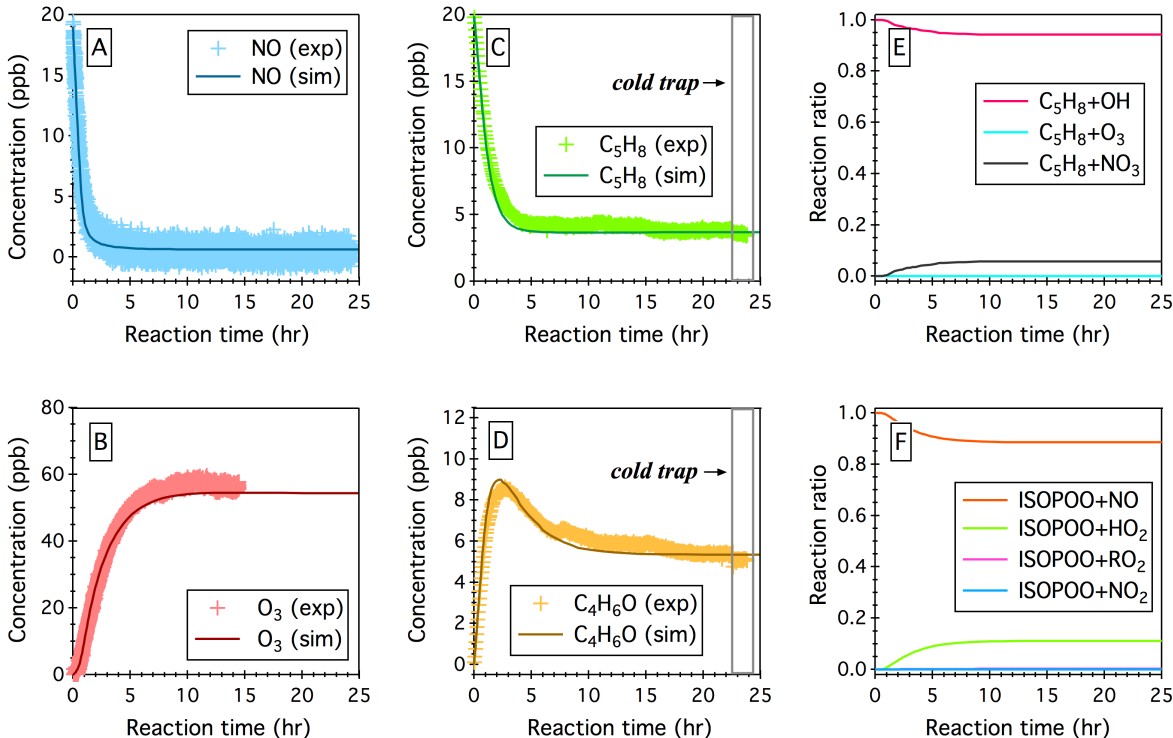

Figure 3. Simulated (sim.) and observed (exp.) temporal profiles of (A) NO, (B) $O_3$, (C) isoprene ($C_5H_8$), and (D) the sum of MACR and MVK ($C_4H_6O$) over 24 hours OH-initiated oxidation of isoprene in the continuous-flow mode chamber operation. Also displayed here include (E) simulated fractions of OH-oxidation, ozonolysis, and $NO_3$-oxidation as the removal pathways of isoprene, and (F) simulated fractions of ISOPOO radicals that react with NO, $HO_2$, $RO_2$, and $NO_3$. Time 0 is the point at which the chamber lights are turned on. Initial experimental conditions are 19 ppb NO, 0 ppb $NO_2$, 0 ppb $O_3$, 600 ppb $H_2O_2$, and 19.9 ppb $C_5H_8$, with continuous input of 600 ppb $H_2O_2$, 19 ppb NO, and 19.9 ppb $C_5H_8$ over the course of 24 hour photochemical reactions.

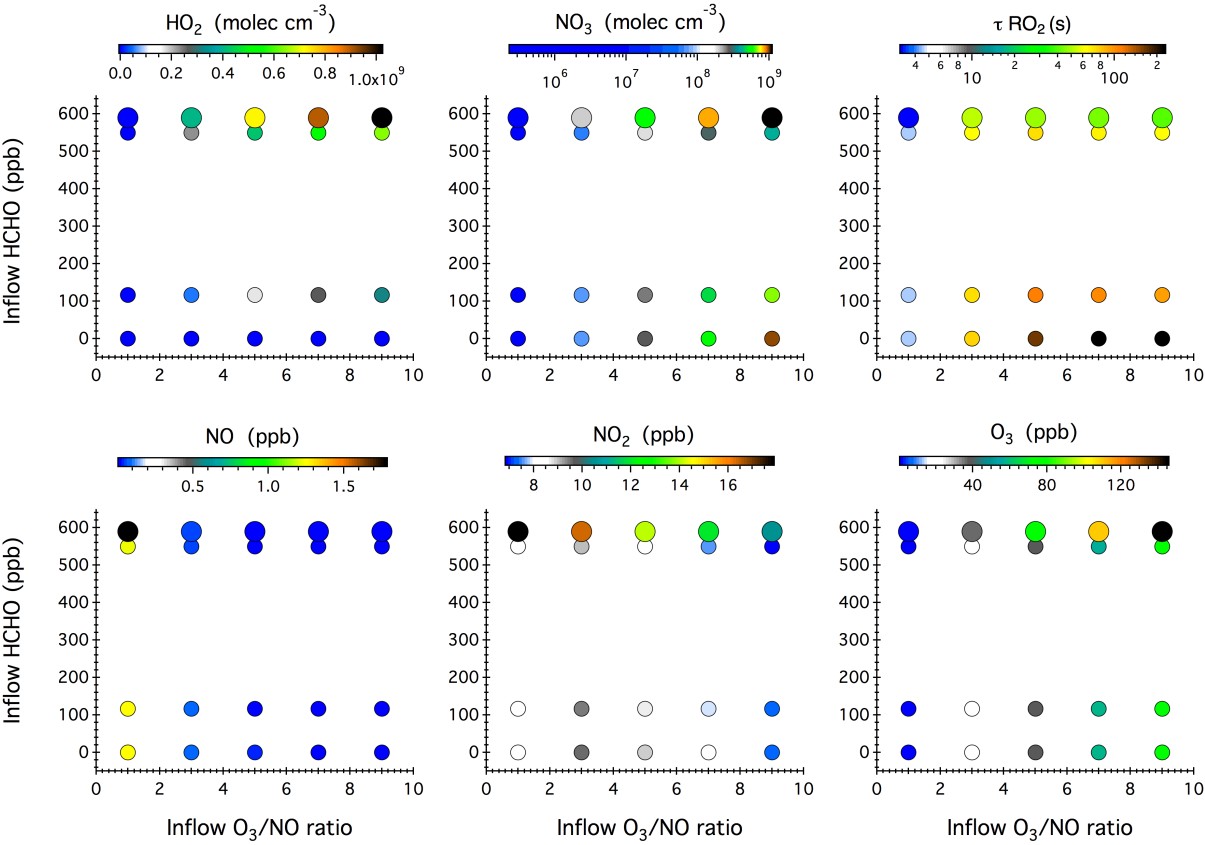

Figure 4. Simulated steady-state mixing ratios of $HO_2$, $NO_3$, NO, $NO_2$, and $O_3$ after 16 hours of dark reactions in the chamber as a function of the concentrations of NO and $O_3$ in the continuous injection flow. The symbol size denotes different inflow NO concentrations, i.e., 10 ppb and 20 ppb. Also given here is the calculated lifetime of an $RO_2$ radical ($\tau_{RO_2}$) with respect to reactions with NO, $NO_3$, and $HO_2$.

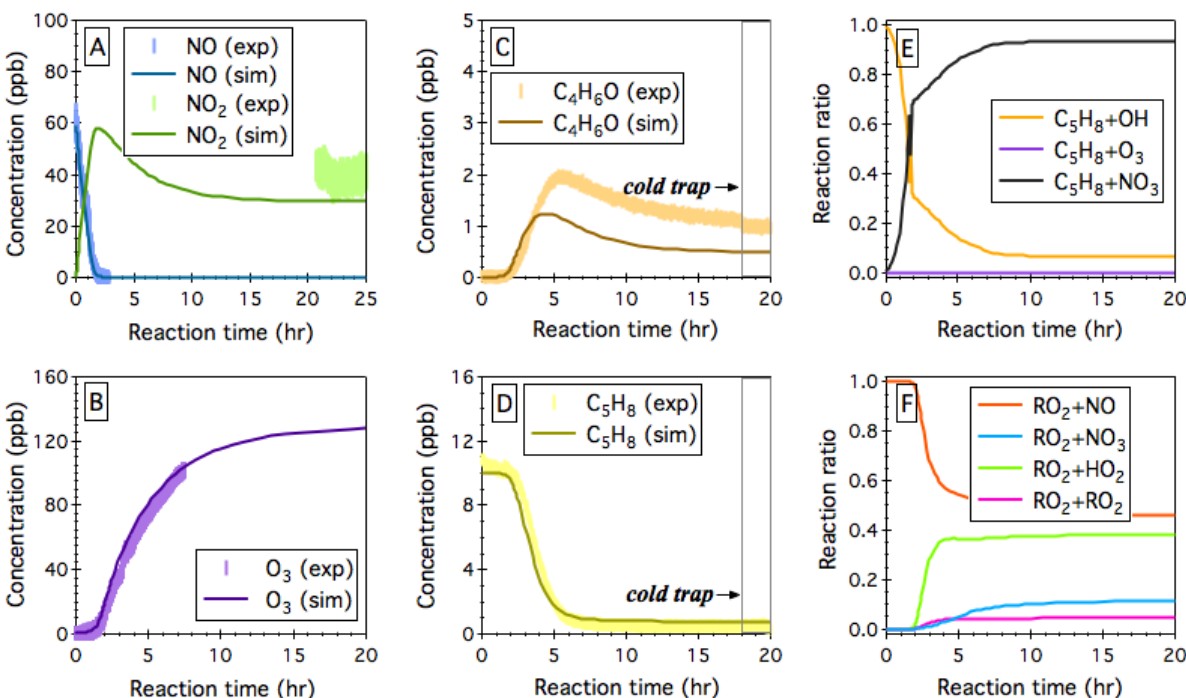

Figure 5. Simulated (sim.) and observed (exp.) evolution patterns of (A) $NO_x$, (B) $O_3$, (C) the sum of MACR and MVK ($C_4H_6O$), and (D) isoprene ($C_5H_8$) over 25 hours $NO_3$-initiated oxidation of isoprene under continuous-flow mode chamber operation. The fractions of isoprene that react with OH, $O_3$, and $NO_3$ are given in panel (E), and the fractions of INOO radical that undergo bimolecular reactions with NO, $NO_3$, $HO_2$, and $RO_2$ are given in panel (F). Initial experimental conditions are 0 ppb $O_3$, 59 ppb $NO_x$, and 10.2 ppb $C_5H_8$, with continuous input of 205 ppb $O_3$, 59 ppb NO, and 10.2 ppb $C_5H_8$ over the course of 25 hour dark reactions.