# Peer review of "A Steady State Continuous Flow Chamber for the Study of Daytime and Nighttime Chemistry under Atmospherically Relevant NO levels"

_Atmospheric Measurement Techniques, 2018_

## Referee Comment (RC1) · Anonymous Referee #1 · 29 Jan 2018

The authors report a systematic investigation of the conditions achieved in a 10 m^3 simulation chamber and discuss this in the context of isoprene oxidation experiments. The manuscript is well written and within the scope of the journal. There are only few minor points which should be clarified before publication.

The authors could add a short comparison of their conditions with conditions in other chambers, which also work at atmospheric conditions, in addition to emphasizing that previous studies worked at either zero or high NO conditions.

P5: Some details about the chamber air supply could be added to justify how the low range of NOx is achieved. The authors state that the minimum relative humidity in the chamber is 10%. Is this limited by the purification process?

P7 l204: I assume that the authors mean that MVK and MACR cannot be distinguished by a PTR instrument because of their same mass and not because of the same detection sensitivity.

P7: How was avoided that frozen water in the trap in the inlet of the PTR instrument disturbs measurements?

P8/P11: The authors assume that there are no wall loss effects. Does this also apply to OH, HO2, NO3, O3 shown in for example Fig. 1? How does this compare to findings in other chambers?

P10 l332: There is another study investigating the MVK and MACR yields at similar conditions that the authors may want to add (Karl et al., J Atmos Chem 55, 167-185, 2006).

P10 l345: What about photolysis and ozonolysis reactions of product species? Please quantify, if they contributed to the loss of these species.

---

## Referee Comment (RC2) · P. Seakins (Referee) · 29 Jan 2018

Studying atmospheric processes such as isoprene oxidation under realistic concentrations of NOx, hydrocarbons and radicals is an important goal in chamber studies. Improvements in VOC measurement techniques allow these compounds to measured at low concentrations with a resulting decrease in the radical concentrations required to detect measureable differences in concentrations and in general to operate at concentrations much closer to those of ambient conditions, providing a better test of chemical models. However, working at ambient concentrations extends the duration of the experiment; this has a practical effect on the number of repetitions, but potentially increases the role of heterogeneous chemistry.

The authors present initial data from a new steady-state, slow continuous flow chamber looking at background chemistry and reporting some initial results on isoprene oxidation. I think it would be helpful for the authors to address the following points to improve what is already a good paper.

1) As Ref #1 has already commented, it would be helpful to provide some comparisons with other chambers which are capable of operating under zero to medium NOx conditions.

2) AMT is a technical journal and so I think it would be appropriate to include some more technical aspects (e.g. spectra of the lamps, temperature profiles across the chamber when lamps in operation, rationale for minimum 10% rel humidity - presumably it would be possible to run with cylinder air if necessary). Is the air in the chamber mechanically mixed or just relies on incoming air flow? How was the mixing time determined?

3) An important aspect of any simulation chamber and particularly one with long reaction times is the reproducibility of the results. Have repeat measurements been carried out? When working with higher concentrations of NOx is there any evidence of wall reactions or that the walls can be a source of HOx (HONO)?

4) Isoprene chemistry is a topical and very important subject, but, as the authors point out, it is a system where there is still some uncertainty in the chemistry. Have the authors carried out any intermediate studies (e.g. ethane or butane oxidation) where the chemistry is better defined. Reproducing results from a simpler VOC system would give greater confidence that the excellent data obtained for isoprene can be directly compared with the literature.

---

## Author Comment (AC1) · 4 Apr 2018

**Response to Reviewer #1**

The authors report a systematic investigation of the conditions achieved in a 10 m$^3$ simulation chamber and discuss this in the context of isoprene oxidation experiments. The manuscript is well written and within the scope of the journal. There are only few minor points which should be clarified before publication.

*We thank reviewer #1 for the constructive and insightful comments. Our point-by-point responses can be found below, with reviewer comments in **black**, our responses in **blue**, alongside the relevant revisions to the manuscript in **red**.*

The authors could add a short comparison of their conditions with conditions in other chambers, which also work at atmospheric conditions, in addition to emphasizing that previous studies worked at either zero or high NO conditions.

[Responses] In the revised manuscript, we first introduce four experimental methods that have been used in previous studies targeting at a controlled NO level from a few hundreds of ppt to a few ppb. Then in Section 4.1, we add a paragraph comparing the photochemical oxidation environment created in the present study with the 'intermediate-NO' conditions achieved by chambers that employed these experimental approaches.

[Revisions] We have added following discussions in the revised manuscript:

"Experimental approaches targeting at a controlled NO level (sub-ppb to ppb) have been introduced over the years. For outdoor chambers, experiments were typically performed by exposing a gas mixture of $O_3/NO_x/VOCs$ or $HONO/NO_x/VOCs$ to natural sunlight (Bloss et al., 2005; Karl et al., 2006). OH radicals were produced either via the photolysis of ozone and subsequent reaction of $O(^1D)$ with $H_2O$ or directly from the photolysis of HONO. NO levels ranging from a few hundreds of ppt to a few ppb over the course of several hours of reactions have been reported. In the absence of any additional supply, NO will be eventually depleted in a closed chamber environment, and the initial 'moderate-NO' condition will essentially transfer to the 'zero-NO' condition. For indoor chambers, a 'slow chemistry' scenario initiated by photolyzing methyl nitrite ($CH_3ONO$) under extremely low UV intensities as the OH radical source ($J_{CH3ONO} \sim 10^{-5}$ s$^{-1}$) was created to study the autoxidation chemistry of peroxy radicals produced from isoprene photooxidation (Crounse et al., 2011; Crounse et al., 2012; Teng et al., 2017). The resulting NO and $HO_2$ mixing ratios are maintained at ~ppt level ($CH_3ONO + O_2 + h\nu \rightarrow HO_2 + NO + HCHO$) over the course of several hours of reaction, and the average OH concentration (OH

~ $10^5$ molec cm$^{-3}$) is approximately one order of magnitude lower than that in the typical daytime ambient atmosphere. Another example relates to a recent method development in the Potential Aerosol Mass (PAM) flow tube reactor where nitrous oxide (N$_2$O) was used to produce ~ppb level of NO (O$_3$ + $hv$ → O$_2$ + O($^1$D); O($^1$D) + N$_2$O → 2NO) (Lambe et al., 2017). Timescales for chemical reactions and gas-particle partitioning are ultimately limited to the mean residence time (~80 s) of the PAM reactor."

"For example, a steady-state NO level at ~1 ppb was created by the continuously mixed flow chamber operation for the study of isoprene photooxidation chemistry (Liu et al., 2013)."

"We further compare the photochemical oxidation environment created here with the 'intermediate-NO' conditions achieved by other chambers that employed the experimental approaches introduced earlier. In terms of the oxidizing power, all approaches are capable of maintaining an atmospheric relevant OH level (~$10^6$ molec cm$^{-3}$), expect the 'slow chemistry' scenario that limits the photolysis rate of the OH precursor and results in an average OH mixing ratio of ~$10^5$ molec cm$^{-3}$ (Crounse et al., 2012; Teng et al., 2017). At comparable OH levels, the overall atmospheric OH exposure achieved in the flow tube reactor is rather limited due to the short residence time (e.g., ~80 s in the PAM reactor). In terms of the NO$_x$ level, precisely controlled steady-state NO concentration can be achieved for an indefinite time period by operating chambers in the continuously mixed flow mode. However, NO$_2$ accumulates during the continuous oxidation process and the resulting NO$_2$/NO ratio can be as much as an order of magnitude higher than that achieved in the static outdoor chambers."

P5: Some details about the chamber air supply could be added to justify how the low range of NOx is achieved. The authors state that the minimum relative humidity in the chamber is 10%. Is this limited by the purification process?

[Responses] We have performed a series of experiments investigating the isoprene daytime and nighttime chemistry under a wide range of steady-state NO levels (from ~30 ppt to ~300 ppb). For these experiments, we used a concentrated NO cylinder (NO = 133.16 ppm, balance N$_2$) upon dilution by purified air to constantly flow a desired concentration of NO into the chamber. We used a high sensitivity chemiluminescence instrument with a detection limit of ~ 25 ppt to measure the NO concentrations. The lowest steady-state NO concentration achieved in the chamber is around 30 pptv. Detailed results will be forthcoming in a future publication.

We use an AADCO zero air generator (Model 737-42) to purify the house air. This generator contains a methane burner, which is the main source of water vapor in the chamber flushing air (RH ≤ 10%). We have added a new figure in the Supplement showing the temperature and relative

humidity temporal profiles during a typical ~20 h continuous flow experiment. The RH started at ~9%, rapidly decreased after lights on, and finally stabilized at ~5% at ~306 K.

[Revisions] We have added following sentences in the revised manuscript:

"Constant NO injection flow was achieved by diluting the gas flow from a concentrated NO cylinder (NO = 133.16 ppm, balance $N_2$) to a desired mixing ratio (0.1–100 ppb) using a set of mass flow controllers (Tylan FC260 and FC262, Mykrolis Corp., MA). The lowest steady-state NO level that can be achieved in the chamber is around 30 pptv (unpublished, NCAR)."

"The relative humidity of the chamber air is below 10% under dry conditions (the remaining water vapor is generated from methane combustion during the air purification process)…"

P7 l204: I assume that the authors mean that MVK and MACR cannot be distinguished by a PTR instrument because of their same mass and not because of the same detection sensitivity.

[Responses] Yes, MVK and MACR are both detected as ion $C_4H_7O^+$ ($m/z$ 71) by the proton-transfer-reaction ionization scheme. If the instrument sensitivity towards MACR was different from that of MVK, then it would become impossible to calculate their total concentration based on the measured $C_4H_7O^+$ ion intensity, because the relative molar ratio of MACR to MVK produced from the isoprene photooxidation is unknown. When we performed calibration for each compound, we found that the instrument sensitivity towards each of them is the same so that we could use one calibration factor to retrieve the sum of MACR and MVK concentration based on the measured total $C_4H_7O^+$ ion intensity.

[Revisions] We have revised the main text:

"…that is, isoprene is detected as ion $C_5H_9^+$ ($m/z$ 69) and MACR and MVK are both detected as ion $C_4H_7O^+$ ($m/z$ 71)."

"…and as a result, the sum of MACR and MVK concentration in the sampling air can be calculated by applying one calibration factor to the measured $C_4H_7O^+$ ($m/z$ 71) ion intensity."

P7: How was avoided that frozen water in the trap in the inlet of the PTR instrument disturbs measurements?

[Responses] We have found that the $H_3O^+$ ($m/z$ 19) reagent ion intensity, together with all the measured analyte ion intensities, decreased upon submerging the sampling tubing in the low temperature (−40±2 °C) ethanol bath. The decrease in the measured analyte ion signals is simply due to the fact that less water is available for the ionization process, but this would not impact the

measured concentrations of isoprene, MACR, and MVK, as the intensity of each ion of interest (counts per second) is simultaneously normalized to the total $H_3O^+$ reagent ion intensity. We have confirmed experimentally that the normalized analyte ion intensities (and thus the calculated concentrations) of isoprene, MACR, and MVK from a mixture of standard gases remain the same under cold-trapping conditions, suggesting no disturbance of the cold trap procedure.

P8/P11: The authors assume that there are no wall loss effects. Does this also apply to OH, $HO_2$, $NO_3$, $O_3$ shown in for example Fig. 1? How does this compare to findings in other chambers?

[Responses] Observations from previous and recent chamber studies (McMurry and Grosjean et al., 1985; Matsunaga and Ziemann, 2010; Zhang et al., 2015; Krechmer et al., 2016; Ye et al., 2016; Huang et al., 2018) have reached a consensus that under dry conditions, the chamber wall induced deposition rate and the fraction that remains in the chamber walls of organic vapors largely depend on the volatility of the organic vapors. For volatile organic compounds like isoprene, MACR, and MVK, gas-wall partitioning equilibrium can be established within a few seconds and minimal wall losses have been observed. Therefore it is valid to assume 'no wall loss effects' for volatile organic compounds under dry conditions employed in the present study.

We did not observe any ozone dark loss on the Teflon chamber wall over a period of several hours in the NCAR chamber. We found a study by Wang et al. (2014) showing that the measured ozone wall loss rate in a 30 $m^3$ Teflon chamber is ~$2.18\times10^{-6}$ $s^{-1}$. This loss rate is much lower compared with the other ozone removal processes in the chamber, e.g., reactions with $NO_x$, and thus can be neglected in modeling the steady-state ozone concentration.

It is currently unclear if the Teflon chamber wall acts as a source or sink of free radicals. Carter et al. (1982) suggested that the chamber wall might be a potential source of OH radicals, and the radical input rates depend on the light intensity and the type of chamber employed. On the other hand, heterogeneous chemistry occurring on the wall, e.g., cyclization and dehydration of δ-hydroxycarbonyl to substituted dihydrofuran (Lim and Ziemann, 2009), might consume a certain amount of OH radicals. In the absence of direct measurements of the wall-induced free radical generation or consumption rates, we are currently unable to represent this process in the model.

[Revisions] We have stated in Page 7:

"Note that two terms are neglected in Equation (1), i.e., organic vapor condensation onto particles and deposition on the chamber wall. This is a reasonable simplification here owing to the relatively high volatility (≥ $10^{-1}$ atm) of compounds studied (Matsunaga and Ziemann, 2010; Zhang et al., 2015; Krechmer et al., 2016; Huang et al., 2018)."

P10 l332: There is another study investigating the MVK and MACR yields at similar conditions that the authors may want to add (Karl et al., J Atmos Chem 55, 167-185, 2006).

[Responses] Thanks for providing this reference, which has been added in the revised manuscript.

[Revisions] We have added the following sentence in the revised manuscript:

"Measurements by Karl et al. (2006) and Liu et al. (2013) conducted at NO concentrations comparable to the moderately polluted urban environment (~ 0.2 ppb in Karl et al. and ~1 ppb in Liu et al.) found higher MACR (~27% in Karl et al. and ~31.8% in Liu et al.) and MVK (~41% in Karl et al. and ~44.5% in Liu et al.) yields than other studies."

P10 l345: What about photolysis and ozonolysis reactions of product species? Please quantify, if they contributed to the loss of these species.

[Responses] At steady state, the measured/simulated ozone level is ~55 ppb, and the simulated OH concentration is approximately $2.6 \times 10^6$ molecules $cm^{-3}$. The photolysis rates of MACR and MVK under the maximum irradiation conditions employed in this study are calculated as $1.25 \times 10^{-7}$ $s^{-1}$ and $1.55 \times 10^{-7}$ $s^{-1}$, respectively. The reaction rate constants for MACR+OH, MACR+$O_3$, MVK+OH, and MVK+$O_3$ at 306 K are $2.77 \times 10^{-11}$ $cm^3$ $molec^{-1}$ $s^{-1}$, $1.46 \times 10^{-18}$ $cm^3$ $molec^{-1}$ $s^{-1}$, $1.91 \times 10^{-11}$ $cm^3$ $molec^{-1}$ $s^{-1}$, and $5.92 \times 10^{-18}$ $cm^3$ $molec^{-1}$ $s^{-1}$, respectively. As we use $C_4H_6O$ to represent the total amount of MACR and MVK, the photolysis rate of $C_4H_6O$ and the reaction rate constants for $C_4H_6O$+OH and $C_4H_6O$+$O_3$ are taken as the average, i.e., $1.40 \times 10^{-7}$ $s^{-1}$, $2.34 \times 10^{-11}$ $cm^3$ $molec^{-1}$ $s^{-1}$, and $3.69 \times 10^{-18}$ $cm^3$ $molec^{-1}$ $s^{-1}$, respectively. We then calculate the fractions of $C_4H_6O$ that undergo reactions with OH, $O_3$, and photolysis are 94.0%, 5.8%, and 0.2%, respectively. As the ozonolysis and photolysis in total account for 6% of the $C_4H_6O$ degradation, we have decided to neglect these two pathways in the calculation.

[Revisions] We have added the following sentence in the main text:

"The ozonolysis and photolysis in total account for ~6% of the $C_4H_6O$ degradation pathway and are neglected here as well."

---

## Author Comment (AC2) · 4 Apr 2018

**Response to Prof. Paul Seakins**

Studying atmospheric processes such as isoprene oxidation under realistic concentrations of NOx, hydrocarbons and radicals is an important goal in chamber studies. Improvements in VOC measurement techniques allow these compounds to be measured at low concentrations with a resulting decrease in the radical concentrations required to detect measureable differences in concentrations and in general to operate at concentrations much closer to those of ambient conditions, providing a better test of chemical models. However, working at ambient concentrations extends the duration of the experiment; this has a practical effect on the number of repetitions, but potentially increases the role of heterogeneous chemistry. The authors present initial data from a new steady-state, slow continuous flow chamber looking at background chemistry and reporting some initial results on isoprene oxidation. I think it would be helpful for the authors to address the following points to improve what is already a good paper.

*We thank Prof. Paul Seakins for the constructive and insightful comments. Our point-by-point responses can be found below, with reviewer comments in **black**, our responses in **blue**, alongside the relevant revisions to the manuscript in **red**.*

1) As Ref #1 has already commented, it would be helpful to provide some comparisons with other chambers which are capable of operating under zero to medium $NO_x$ conditions.

[Responses] In the revised manuscript, we first introduce four experimental methods that have been used in previous studies targeting at a controlled NO level from a few hundreds of ppt to a few ppb. Then in Section 4.1, we add a paragraph comparing the photochemical oxidation environment created in the present study with the 'intermediate-NO' conditions achieved by chambers that employed these experimental approaches.

[Revisions] We have added following discussions in the revised manuscript:

"Experimental approaches targeting at a controlled NO level (sub-ppb to ppb) have been introduced over the years. For outdoor chambers, experiments were typically performed by exposing a gas mixture of $O_3/NO_x/VOCs$ or $HONO/NO_x/VOCs$ to natural sunlight (Bloss et al., 2005; Karl et al., 2006). OH radicals were produced either via the photolysis of ozone and subsequent reaction of $O(^1D)$ with $H_2O$ or directly from the photolysis of HONO. NO levels ranging from a few hundreds of ppt to a few ppb over the course of several hours of reactions have been reported. In the absence of any additional supply, NO will be eventually depleted in a closed chamber environment, and the initial 'moderate-NO' condition will essentially transfer to

the 'zero-NO' condition. For indoor chambers, a 'slow chemistry' scenario initiated by photolyzing methyl nitrite ($CH_3ONO$) under extremely low UV intensities as the OH radical source ($J_{CH3ONO} \sim 10^{-5}$ s$^{-1}$) was created to study the autoxidation chemistry of peroxy radicals produced from isoprene photooxidation (Crounse et al., 2011; Crounse et al., 2012; Teng et al., 2017). The resulting NO and $HO_2$ mixing ratios are maintained at ~ppt level ($CH_3ONO + O_2 + hv \rightarrow HO_2 + NO + HCHO$) over the course of several hours of reaction, and the average OH concentration (OH ~ $10^5$ molec cm$^{-3}$) is approximately one order of magnitude lower than that in the typical daytime ambient atmosphere. Another example relates to a recent method development in the Potential Aerosol Mass (PAM) flow tube reactor where nitrous oxide ($N_2O$) was used to produce ~ppb level of NO ($O_3 + hv \rightarrow O_2 + O(^1D)$; $O(^1D) + N_2O \rightarrow 2NO$) (Lambe et al., 2017). Timescales for chemical reactions and gas-particle partitioning are ultimately limited to the mean residence time (~80 s) of the PAM reactor."

"For example, a steady-state NO level at ~1 ppb was created by the continuously mixed flow chamber operation for the study of isoprene photooxidation chemistry (Liu et al., 2013)."

"We further compare the photochemical oxidation environment created here with the 'intermediate-NO' conditions achieved by other chambers that employed the experimental approaches introduced earlier. In terms of the oxidizing power, all approaches are capable of maintaining an atmospheric relevant OH level (~$10^6$ molec cm$^{-3}$), expect the 'slow chemistry' scenario that limits the photolysis rate of the OH precursor and results in an average OH mixing ratio of ~$10^5$ molec cm$^{-3}$ (Crounse et al., 2012; Teng et al., 2017). At comparable OH levels, the overall atmospheric OH exposure achieved in the flow tube reactor is rather limited due to the short residence time (e.g., ~80 s in the PAM reactor). In terms of the $NO_x$ level, precisely controlled steady-state NO concentration can be achieved for an indefinite time period by operating chambers in the continuously mixed flow mode. However, $NO_2$ accumulates during the continuous oxidation process and the resulting $NO_2$/NO ratio can be as much as an order of magnitude higher than that achieved in the static outdoor chambers."

2) AMT is a technical journal and so I think it would be appropriate to include some more technical aspects (e.g. spectra of the lamps, temperature profiles across the chamber when lamps in operation, rationale for minimum 10% relative humidity – presumably it would be possible to run with cylinder air if necessary). Is the air in the chamber mechanically mixed or just relies on incoming air flow? How was the mixing time determined?

[Responses] We have added a new figure showing the spectra of the 100% UV lamps used in the experiment as well as the temperature and relative humidity profiles across the chamber

during a 20 h continuous flow experiment under maximum radiation conditions in the supplement, also given below.

As we constantly flowing 40 L/min purified dry air into the chamber during the continuous flow mode operation, it is financially impractical to use cylinders as the air supply. We use an AADCO zero air generator (Model 737-42) to purify the house air. This generator contains a methane burner, which is the main source of water vapor in the chamber flushing air (RH ≤ 10%). As shown in Figure S1, RH started at ~9% at the beginning of the experiment and dropped to ~5% when temperature increased to ~ 307 K.

Chamber mixing relies on the flushing air at a constant flow rate of 40 L/min. A tracer compound, e.g., $CO_2$ and NO, is injected into the chamber. The mixing time is then established by the time it takes for the measurement signal to stabilize following injection. A combination of different injection and sampling ports are used to determine the average mixing time.

[Revisions] Figure S1 showing the spectra of 100% UV lights and T/RH temporal profiles during a 20 h continuous flow experiment is given below:

[Figure]

We have added following discussions in the main text:

"The relative humidity of the chamber air is below 10% under dry conditions (the remaining water vapor is generated from methane combustion during the air purification process)…"

"Typical temperature and relative humidity profiles across the chamber under maximum irradiation conditions are given in Figure S1 in the Supplement."

"The chamber is actively mixed by the turbulence created by the 40 L min-1 flushing air. The characteristic mixing time is defined as the time it takes for the measurement signal of a tracer

compound (e.g., $CO_2$ and NO) to stabilize following a pulse injection. The average mixing time in the NCAR chamber was determined to be ~9 min, which is ~4% of the residence time."

3) An important aspect of any simulation chamber and particularly one with long reaction times is the reproducibility of the results. Have repeat measurements been carried out? When working with higher concentrations of $NO_x$ is there any evidence of wall reactions or that the walls can be a source of $HO_x$ (HONO)?

[Responses] We have performed a series of experiments to examine the reaction mechanisms for the OH-initiated oxidation of isoprene under a wide range of steady state NO levels, i.e., from ~80 ppt to ~300 ppb. The molar yields of MACR and MVK were measured by GC-FID. The measured molar yields of MACR and MVK from two replicate experiments agree within 4.4% and 1.7%, respectively. We are currently preparing these results for a future publication.

We found that the chamber wall might be a potential source of $NO_x$ under the so-called 'low NO conditions'. We ran a blank experiment by flushing the chamber for > 24 h, keeping the chamber at static mode under dark conditions for an hour, and then turning 100% UV lights on for another few hours. We used a high sensitivity chemiluminescence instrument with a detection limit of ~ 25 ppt to measure the NO concentration in the chamber. When the chamber was operated at static mode under dark conditions, we did not observe any NO in the chamber, indicating zero penetration of room air into the Teflon chamber. When we turned on the UV lights, we saw a sharp increase of NO by about 30-40 ppt. This amount of NO stayed in the bag during lights on and gradually diminished after lights off. We suspect that the observed NO was produced from the photolysis of HONO deposited on the Teflon wall.

4) Isoprene chemistry is a topical and very important subject, but, as the authors point out, it is a system where there is still some uncertainty in the chemistry. Have the authors carried out any intermediate studies (e.g. ethane or butane oxidation) where the chemistry is better defined. Reproducing results from a simpler VOC system would give greater confidence that the excellent data obtained for isoprene can be directly compared with the literature.

[Responses] Isoprene chemistry under high NO conditions has been extensively studied. The molar yields determined from previous studies range from 30–35% for MVK and 20–25% for MACR in the presence of > 60 ppb NO (Tuazon and Atkinson, 1990; Paulson and Seinfeld, 1992; Miyoshi et al., 1994; Ruppert and Becker, 2000; Sprengnether et al., 2002; Galloway et al.,

2011). We measured the molar yields of MACR and MVK from isoprene photooxidation in the presence of ~ 300 ppb NO using GC-FID. Our results, 24.9% for MACR and 32.2% for MVK, fall within the range obtained by previous measurements.

We investigated the OH-initiated oxidation of butane and butene under both high and low NO conditions. Experiments were performed in the static mode and typically lasted for several hours. The goal of these experiments is to examine the possibility and mechanisms of thermal decomposition of fragile products in the GC and PTRMS sampling system. We measured the molar yields of small carbonyl products, such as propanal and MEK, with the use of a cold trap system in front of the instrument inlets. Our measurements agree well with the predictions by the MCM mechanism. Detailed results will be forthcoming in a future publication.